# Knowledge, Attitudes, and Practices of Communal Livestock Farmers regarding Animal Health and Zoonoses in Far Northern KwaZulu-Natal, South Africa

**DOI:** 10.3390/ijerph20010511

**Published:** 2022-12-28

**Authors:** Yusuf Bitrus Ngoshe, Eric Etter, Jose Pablo Gomez-Vazquez, Peter N. Thompson

**Affiliations:** 1Epidemiology Section, Department of Production Animal Studies, Faculty of Veterinary Science, University of Pretoria, Private Bag X04, Onderstepoort 0110, South Africa; 2ASTRE, University of Montpellier, CIRAD, INRA, F-34000 Montpellier, France; 3CIRAD, UMR Animal, Santé, Territoires, Risque et Ecosystèmes (ASTRE), F-97170 Petit-Bourg, France; 4Center for Disease Modelling and Surveillance (CADMS), University of California, Davis, CA 95616, USA

**Keywords:** KAP, communal livestock farmers, animal health, zoonoses, livestock-wildlife interface

## Abstract

The presence of zoonotic diseases adversely affects livestock production and farmers’ livelihood in communal areas. A lack of awareness about zoonotic diseases among rural farmers results in economic losses and health risks. The far north-eastern corner of KwaZulu-Natal (KZN) province, South Africa, is home to large numbers of communal livestock farmers who live adjacent to wildlife reserves and international borders. There have been reports of zoonotic and trade-sensitive diseases in the area, but farmers’ knowledge, attitudes, and practices (KAP) in this regard are poorly understood. This study investigated the KAPs of communal livestock farmers in far northern KZN regarding livestock and zoonotic diseases found in the livestock–wildlife interface and determined the constraints and challenges faced by communal livestock farmers. A cross-sectional questionnaire survey was conducted among 504 livestock farmers at 45 dip tanks between August and November 2020, using a closed-ended questionnaire. Although the overall level of knowledge regarding animal disease transmission was fairly good (score: 53.2%), 25.4% and 21.4% of farmers had moderate and poor knowledge, respectively, about zoonotic disease transmission and prevention. Over 40% of the farmers were not aware of the zoonotic nature of wildlife and livestock diseases. Older farmers, despite their lower level of education, were more knowledgeable on animal diseases and had better practices in regard to zoonotic disease prevention and management compared to younger ones. The majority of farmers cited the lack of water, insufficient grazing land, stock theft, the restriction of animal movement, and animal diseases as the most significant challenges they faced regarding animal production. The results indicate the need for extension programs that target educating livestock farmers to improve their knowledge of these diseases.

## 1. Introduction

Livestock is one of the major contributors to poverty alleviation, food security, and sustainable livelihoods in both rural [1] and urban areas [2]. However, diseases and parasites are serious constraints to rural livestock production in communal areas and where livestock-wildlife interactions are common [3]. Small-scale farmers rely on their animals for food, milk, and a living, but when disease threatens a herd or flock, the health and welfare of a farmer and the community are jeopardized and animal diseases are an ongoing food security problem [4]. Reduced growth rate, lower quantity and quality of market outputs, increased preventative, treatment and control expenses, and risks to human health are some of the consequences of livestock diseases [5,6].

About 60% of pathogens that cause human diseases are of animal origin [7]. Zoonotic pathogens threaten the lives and livelihoods of people that depend on livestock as a source of food and income [8]; however, there is a lack of awareness about zoonotic diseases among rural livestock farmers worldwide [9]. 

In South Africa, there are two types of livestock farming sectors: the commercial and the subsistence farming systems. The latter is commonly practiced by communal rural farmers and contributes approximately 40% to agricultural income [10,11]. Cattle and goats are the most widely reared livestock and one of the major economic activities in the far northern part of KwaZulu-Natal (KZN), and they contribute significantly to food security and the national economy. In KZN, the sale of goats and cattle is worth about R4.2 billion per annum [12]. However, livestock in this region is more than just a source of food and milk, but also reflects a person’s wealth and is used to pay lobola (dowries) and household needs such as school fees, and is a source of transportation, draught power and traditional clothing [13].

Far north-eastern KZN shares borders with Eswatini and Mozambique and is an area of special concern due to the presence of, or potential for introduction and spread of, transboundary animal diseases, including trade-sensitive diseases such as Rift Valley fever (RVF) [14], foot-and-mouth disease (FMD) [15,16], and African swine fever [17]. However, the knowledge, attitudes, and practices (KAP) of farmers in the area regarding livestock and zoonotic diseases are poorly understood.

The aim of this study was to assess the KAP of communal livestock farmers in far northern KwaZulu-Natal in regard to livestock diseases. The specific objectives were to (1) identify and rank disease syndromes that farmers consider important, (2) determine farmers’ perceptions of the level of disease occurrence in the study area, (3) determine if farmers consider animal disease as a potential threat to livestock health and their livelihood, (4) assess the knowledge of the farmers about zoonotic diseases, and (5) identify the constraints and challenges faced by communal livestock farmers in the study area.

## 2. Materials and Methods

### 2.1. Ethical Considerations

Ethical approvals were obtained from the Research Ethics Committees of the Faculty of Veterinary Science and the Faculty of Humanities of the University of Pretoria (REC151-19). Approvals were also obtained from the local *izinkosi* (kings or chiefs) before the project began and signed informed consent was obtained from each respondent before their participation in the survey. 

### 2.2. Study Area

This study was conducted in two local municipalities; Jozini (Latitude: −27.429366, Longitude: 32.065107) and uMhlabuyalingana (Latitude: −27.1894900, Longitude: 32.5612000) of the Umkhanyakude District in the far north-eastern part of KZN province. It has a hot and humid tropical climate with most of the rainfall falling in summer, between December and March. The study area includes floodplains and pans, with two major rivers, the Phongolo and the Usuthu rivers [18]. The district is among the poorest in the country, with the second-highest socio-economic deprivation index in South Africa [19] and it is estimated that more than 82% of households live below the poverty line [20]. The Zulu ethnic group is predominant in the study area. The indigenous Nguni cattle breed and indigenous “Zulu” goats are the most common livestock in this area [21]. The majority of the farmers rely on livestock rearing on communal land for a living and relatively few practice crop farming.

In southern Africa, the dip tank system is a pool of water with a railing on the side that is used for the control of tick-borne diseases. The pool of water contains a mixture of anti-tick chemicals, and the cattle are drove through, this kills the ticks and other ectoparasites. The dip tank system in rural communities of KZN is run and managed by local livestock associations, with elected chairpersons and secretaries organizing regular cattle dipping days (weekly in summer and fortnightly in winter) in collaboration with the KZN Department of Agriculture and Rural Development (KZNDARD) animal health technicians (AHTs). The dip tanks receive free ectoparasiticide through the AHTs. Dip tanks have an average of 120 farmers, and each one keeps a stock book with a list of the animals they own [12].

The study area is located at a wildlife–livestock interface, with Ndumo Game Reserve and Tembe Elephant Park located in the study area, and borders Eswatini to the west and Mozambique to the north. The area is classified as a controlled zone for FMD, and animal and animal product movements out of, into, within, and through the area are restricted and require movement permits issued by the state veterinary service [22].

### 2.3. Study Design and Sampling Strategy

A cross-sectional survey was conducted between August and November 2020 at the 45 northernmost dip tanks that were the closest to the Mozambique border (Figure 1). The list and locations of the dip tanks, together with the total number of livestock owners per dip tank, were obtained from KZNDARD. At least 5% of the total population of livestock owners at each dip tank (average of 6 persons per dip tank) were selected for the survey. In cases where cattle owners indicated that they also owned goats, they were interviewed for goat-related questions as well. 

### 2.4. Study Participants and Interview Procedure 

Only farmers aged 18 years and above were included in the survey. Meetings were organized through the community leaders and the local AHTs in charge of each dip tank. Depending on the number of farmers attending the meetings, at some dip tanks, respondents were randomly selected while at others all respondents who arrived at the meeting were allowed to participate in the survey. The respondents were interviewed individually and as a group using two separate questionnaires. An individual questionnaire comprising seven sections was printed and a hard copy was used to interview each respondent. 

The participatory epidemiology (PE) approach was used with a different questionnaire for a group survey at each dip tank. Discussions were guided by an open-ended question with each dip tank referred to as a group. The PE tools used were colored print pictures of disease syndromes, a checklist of challenges printed on cardboard, and stones for proportional piling [23]. This approach was chosen to determine whether the opinion of the respondents from the individual survey differs from the PE (group survey). 

The questionnaires were developed in English and translated into isiZulu for delivering to respondents. The questionnaires were pre-tested at one dip tank by 18 farmers that were not included in the study. Three AHTs were involved in reviewing the questionnaires and all recommendations were implemented before the study commenced. All responses were recorded in the English version of the questionnaires. Interviews were conducted in community halls or at the dip tank and each lasted for about 30–40 min. Before each interview session, the dip tank chairperson, or the AHT in charge of the dip tanks, introduced the researcher, who then explained the objectives of the project to the livestock farmers before they were asked if they were willing to sign written informed consent.

### 2.5. Questionnaires

Section (i) of the individual survey questionnaire comprised questions regarding the sociodemographics of the respondents, including age, gender, level of education, principal occupation (livestock rearing, crop farming, employed, i.e., other paid jobs), livestock species owned, and number of each species per household. Sections (ii) to (iv) comprised questions regarding KAPs (see below). Section (v) enquired about sources of animal disease information, section (vi) had questions regarding wildlife-livestock interaction, and section (vii) included questions about the challenges and constraints faced by communal livestock farmers.

A table containing lists of disease syndromes was provided to the farmers and they were asked to select the cell corresponding to the frequency with which they see the disease conditions in their animals ranging from never to every week. Scores were assigned with never = 0, every 6 months = 1, every 3 months = 2, every month = 3, every two weeks = 4, and every week = 5. The frequency of diseases was then ranked using these scores as weights. Another such table was used in the same way to rank the perceived impact of disease conditions on their animals using a four-level Likert scale ranging from 1 (very low) to 4 (very high). All results from individual questionnaires are presented in tables and figures with the exception of Figure 6 and Table 7 representing combined opinions from both individual and PE surveys.

#### 2.5.1. Farmers’ Knowledge regarding Zoonotic Disease Transmission

Eleven knowledge-related questions that focused on the mode of zoonotic disease transmission were delivered to the farmers. The questions were phrased in such a way that the correct answer to each was “yes”, and a farmer was awarded one point for every question he/she answered correctly, with a maximum obtainable score of 11. The level of knowledge was categorized into three levels: low (0–5), moderate (6–8), and high (9–11).

#### 2.5.2. Farmers’ Attitudes: Source of Disease Information, Animal Movement Policy, and Impact of Disease on Financial Income

The respondents were asked to rank the sources of disease information dissemination to them in order of importance. These were measured using a four-level Likert scale ranging from 1 (not at all important) to 4 (very important). The farmers were also asked whether they considered animal disease to be a threat to their financial income and whether they agreed that the current animal movement control policy was the best way to control the spread of FMD.

#### 2.5.3. Farmers’ Practices with Respect to Zoonotic Disease Prevention and Management

Fourteen zoonotic disease-management practice-related questions were delivered to the farmers. The questions were phrased in such a way that the best response regarding disease prevention was “yes/no” for some of the questions. The participants were asked about their exposure to animals and animal products, such as the consumption of raw milk, raw meat, and if they practiced hand washing after touching sick or dead animals. They were also asked about the practice of using protective gear (e.g., gloves, coverall, masks, gumboots) when handling animals and if they practiced prophylactic treatment such as deworming, dipping, and multivitamins to protect their livestock from diseases. 

To understand the respondents’ practices regarding disease prevention, a list of five separate questions was used. They were then asked to select the measure(s) they use for disease prevention from the list. A farmer received 1 point for every measure selected and the points were summed and categorized into three levels: very good practice (>2 points), good (2 points) and poor (0–1 points). 

#### 2.5.4. Challenges Faced by Livestock Farmers 

A list of challenges to livestock farmers was provided in the individual survey questionnaire and the respondents were asked to check all that applied to them and to rank them using a six-point Likert scale ranging from not at all important to extremely important. 

The PE questionnaire included questions on challenges faced and the most common diseases observed by the farmers. For the challenges, the group was asked to identify the challenges they faced and proportional piling was used with 10 stones per individual. The number of stones allocated to each challenge was counted and recorded. For the question on the common diseases observed, each group of farmers were shown pictures of disease syndromes, and they were given 10 stones each to rank the most common syndromes they had observed in their herd (i.e., the most common syndrome received the highest number of stones). Stones allocated to each syndrome were summed as described above. 

### 2.6. Statistical Analysis

The questionnaire data were managed in a Microsoft Excel^®^ spreadsheet before exporting to Stata 14 (StataCorp, College Station, TX, USA) for descriptive analysis, and R statistical software [24] with the packages rpart [25] and randomForest [26] was used for the multivariate analyses. 

To evaluate the association of demographic characteristics and farming practices with the overall knowledge score (the outcome variable), we used a combination of random forest (RF) and linear regression. RF is a non-parametric method based on classification and regression trees (CART), where the data are partitioned into groups that minimize within-group variance and maximize between-group variance [27]. This method is particularly useful when dealing with unbalanced data and exploring relationships on high-dimensional variable spaces (i.e., multiple interactions between a large set of variables). The RF provides a ranking of the influence of the variables on the outcome and how much of the outcome variance is explained by the variables. We used the ranking of variables’ importance obtained from the RF to guide the variable selection to be used for univariate and multivariable regression analysis. The hypothesized causal pathways using the most influential variables are presented in Figure 2. Individual variables were examined first in univariate linear regression models to estimate the average score difference and the association among demographic characteristics and practices, and then we constructed a multivariable model to control for confounding. The variable selection for the multivariable model was done based on the following criteria: the variable has a direct effect on the outcome, or is associated with both the exposure and the outcome and is not on the causal pathway. 

## 3. Results

### 3.1. Sociodemographic Characteristics

In total, 504 livestock farmers from 45 dip tanks participated in the survey. Most of the respondents (48.8%) were between 51–70 years old, very few (6%) were below the age of 25 years, and 4% were above 70 years old. Most of the respondents (82.5%) were males and the principal occupation was livestock rearing (89.7%). Although the majority (68.3%) of the respondents had some form of formal education (Table 1), it was observed that they were unable to read. 

The most common species of livestock reared by the farmers was cattle (96.2%), followed by poultry (82.1%). Goats were owned by 74.8% of farmers, while only 4% owned sheep and 13% owned pigs (Table 2).

### 3.2. Knowledge

Regarding the knowledge of the farmers on zoonotic disease transmission, the question of whether disease can be transmitted via mosquito bites had the highest score of 92.6%, followed by the question of whether animal diseases are preventable (83.5%) and whether one can be infected with disease through the consumption of meat from a dead animal (80.8%). More than half (66.8%) of the farmers knew they can get diseases from touching or consumption of animal blood, but only 59.3% of the farmers knew that they can get a disease from keeping animals adjacent to their houses. Most farmers (78.4%) knew that livestock can be infected with a disease from wildlife, but only 53.4% believed that wildlife can be infected with a disease from livestock (Table 3).

### 3.3. Attitudes

Based on the four-point Likert scale responses of the farmers, AHTs were mentioned by 84.4% of farmers (418/493) as their most important source of information and preferred means of communication, followed by community meetings (52.2%; 238/456) and radio (48.1%; 213/443) (Figure 3). 

### 3.4. Practices

Regarding zoonotic disease transmission, most of the respondents (98.2%; 95% CI: 97.0–99.4) practiced washing hands after touching sick animals, 71% (95% CI: 66.8–75.3) of the respondents reported taking measures such as wearing protective gear to protect themselves when their animals are sick. However, 24.8% (95% CI: 20.9–28.6) of the livestock farmers stated that they eat raw meat and more than half (52.3%; 95% CI: 47.9–56.7) of them reported drinking raw milk. 

The majority (96.8%; 95% CI: 95.3–98.3) of the respondents reported that they obtained movement permits for transporting livestock into or out of their communities. There were 94.6% (95% CI: 92.7–96.6) of respondents that said they report to AHTs when their animals are sick, and 51.5% (95% CI: 47.1–55.9) of the respondents reported that they quarantine new animals before introducing them into their herd. The majority (87.1%; 95% CI: 84.2–90.1) of the respondents practice the slaughter of animals at home. Only 13.1% (95% CI: 10.1–16.0) of the respondent stated that they kraal different livestock species together, while 21.9% (95% CI: 18.2–25.7) stated that they practice kraaling the livestock adjacent to the house (Table 4).

### 3.5. Summary of Farmers’ KAP

A total of 53.2% of the respondents had a high level of knowledge of the mode of disease transmission, followed by 25.4% with a moderate level, and 21.4% had a low level of knowledge (Table 5). Regarding disease prevention, 40.5% (95% CI: 36.3–44.5) of the respondents had poor practice, 25.6% (95% CI: 22.0–29.6) had good practice, while 33.9% (95% CI: 29.9–38.2) practiced very good preventive measures. Half of the respondents (50.4; 95% CI: 46.0–54.8) had good practice when handling sick animals, 46.4% (95% CI: 42.1–50.8) had very good practice, and very few of the respondents (3%; 95% CI: 2.0–5.1) had poor practice. With regard to the disposal of a dead animal, more than half of the respondents (68.9%; 95% CI: 64.7–72.8) had good practice, about a quarter of the respondents (26.6%; 95% CI: 22.9–30.6) had poor practice, and 4.5% (95% CI: 3.0–6.8) of the respondents had good practice. Most of the respondents (91.6%; 95% CI: 89.1–94.0) indicated that animal diseases are a threat to their household financial income. Regarding regulatory control of animal movement due to FMD, 81.6% (95% CI: 78.2–85.0) of the respondents agreed that the animal movement policy is necessary to prevent the spread of FMD (Table 5).

### 3.6. Animal Disease Information

The most frequently seen livestock condition identified by farmers was tick infestation, followed by weakness, weight loss, worms, nasal discharge, and diarrhea. All conditions except sudden death, neonatal mortality, mastitis, abscess, and abortion, were observed by at least 50% of the farmers every week (Figure 4). 

The respondents indicated tick infestation as the disease condition with the highest impact, followed by malnutrition, diarrhea, and abortion. Mastitis was identified as the disease condition with the least impact on their animals (Table 6). 

The relationship between frequency and impact of disease conditions is shown by plotting the weighted frequency scores against the mean Likert scores (Figure 5). In general, the more frequently observed conditions were considered to have the greatest impact, with ticks considered to have the greatest frequency and impact.

To assess the agreement between information obtained from individual and PE group surveys, we use the mean Likert scale from Figure 5 for individual survey and the average proportional piling of stones placed on the colored picture of each disease condition (Table 7). We found out that the opinion of the respondents was similar (Figure 6).

### 3.7. Wildlife-Livestock Interaction

The majority (80.6%; 386/478) of the respondents reported that their livestock did not have contact with wildlife. Of the 94 respondents that reported wildlife contact, 35% (33/94) indicated that their livestock had direct physical contact with wildlife outside game reserves and 28% (27/94) indicated direct physical contact within reserves. Of the latter, 41% were from Bhudlweni, Mabona and Mpala dip tanks, and 11% were from Madlakude dip tank, near the south-western borders of Ndumo Game Reserve, and near the western border of Tembe Elephant Park, respectively. A total of 26% (24/94) indicated that contact occurred indirectly within a space of a football field, and 11% (10/94) said their animals had close contact across the fence of a reserve. Regarding which wildlife species had contact (direct or indirect) with livestock, 53% (50/94) indicated herbivorous species (impala, nyala, red duiker, bushbuck, bushpig, warthog, giraffe, buffalo or elephants), and 47% of the respondents indicated carnivorous species (lions, hyaenas, leopards or wild dogs). Additionally, 21.4% (101/472) of farmers reported that carnivorous species had killed or attacked their livestock in the past year and the remainder indicated that wildlife had never caused any loss or damage to their livestock. 

### 3.8. Challenges Faced by Livestock Farmers

The availability of water was identified by the majority of the livestock keepers as an extremely important challenge, followed by the availability of grazing land, animal theft, restriction of animal movement, animal diseases, and access to veterinary services (Figure 7). Interestingly, animal theft was considered a greater challenge than animal diseases.

### 3.9. Statistical Analysis of Factors Associations

The RF analysis explained 30.92% of the variance in the model prediction of knowledge score, with the three most influential variables being education, age and kraaling livestock adjacent to the house (Figure 8). CART provided an overall representation of the expected animal diseases knowledge score depending on the different combinations of variables. For example, for respondents <50 years old who did not practice disease reporting, the expected score was 4.5; in contrast, for respondents >50 years old, who did not practice kraaling livestock adjacent to the house, agreed with FMD animal movement policies and quarantined new animals, the expected knowledge score was 10 (Figure 8). 

Based on the most influential variables shown in the RF results, we fitted univariable linear regression models to evaluate the expected differences in the animal diseases knowledge score based on demographic characteristics, attitudes, and practices. Results from the univariable models fitted are presented in Table 8. When we included variables that had a direct effect on the knowledge score in a multivariable, we found associations between age and expected knowledge score, older respondents had, on average, higher knowledge scores. On the contrary, for education we found that people with higher education (secondary and tertiary) had lower knowledge scores. Respondents who said that their occupation was livestock rearing had higher scores on average when compared to those whose occupation was crop farming or other employment. The results of the multivariable analysis are presented in Table 9.

The statistical analysis performed explored the relationship between demographic characteristics and farming practices with the scores obtained. Despite having a relatively large sample size (>500 respondents), the distribution of some of the variables collected was quite unbalanced, which represents a challenge when using parametric statistical methods such as regression. Here, we used random forest as an initial exploration of the relationship between the variables and the outcomes and guide the variable selection process for a more in-depth analysis of the expected differences in the score for the variables analyzed. Some of the variables with high relative importance were considered as consequences of the score and were not included in the multivariable regression analysis. Nevertheless, we still explored the expected differences of these variables with the score in the univariate analysis. Our multivariable analysis attempted to control for potential confounding effects with the variables analyzed, but we did not find considerable differences between the estimations made with the univariable analysis and the multivariable model.

## 4. Discussion

This study investigated the KAPs of communal livestock farmers in far northern KwaZulu-Natal in regard to livestock diseases, finding an overall moderate level of knowledge. Older farmers, despite their lower level of education, were more knowledgeable on animal diseases and had better practices in regard to zoonotic disease prevention and management compared to younger farmers. Older farmers were more likely to quarantine new animals before they are introduced into the herd, and they agreed with the implementation of animal movement control to prevent the spread of FMD. Interestingly, we also found that respondents with higher education (secondary and tertiary) had lower animal disease knowledge compared to respondents with little or no formal education. The low level of awareness of animal diseases among respondents with higher education may be the consequence of a lack of interest among youths to participate in livestock farming. Our findings on older farmers’ knowledge are similar to findings in a recent study from Iran that assessed the KAP of small ruminant farmers and found an association between experience and understanding regarding several infectious diseases [28]. Similarly, in a recent study that assessed the risks of zoonotic disease among livestock farmers from smallholder communities in Ethiopia, respondents who never attended school were three times more likely to correctly answer zoonosis-related questions than those who did [29]; in contrast, a recent study that assessed the knowledge and opinion of dairy cattle farmers in Malaysia indicated that those with higher education had a better understanding of zoonoses [30]. 

The knowledge from the farmers about the fact that disease can be transmitted from animals to humans was high, similar to that reported from a study conducted in Erzurum, Turkey [31], and much higher than reports from studies in southern Ethiopia [8], and in West Bengal, India [32]. High knowledge of zoonotic diseases among livestock farming communities living at the wildlife–livestock–human interface in another part of northern KZN South Africa has been reported [33]. However, the farmers’ knowledge of specific aspects of zoonotic disease transmission, such as whether wildlife can be infected by livestock and whether they can be infected by keeping livestock adjacent to their houses was low, this practice was significantly associated with lower knowledge of animal diseases. This lack of knowledge could be attributed to the lack of primary healthcare programs that focus on training farmers on zoonotic diseases and to the fact that AHTs focused mostly on notifiable diseases. Similarly, Refs. [29,33], reported low knowledge of zoonotic diseases transmission in livestock and wildlife among small-scale livestock holders’ community and livestock farmers in KZN, South Africa and Ethiopia, respectively. Therefore, there is a need to provide training to the farmers on locally relevant aspects of disease transmission at the human–livestock–wildlife interface, as this could be attributed to a lack of extension programs that focused on educating the farmers on zoonoses and limited veterinary and health workers in this communities. 

The findings of this study showed a high level of understanding of disease transmission via mosquito bites, which may be a result of ongoing malaria control efforts in the study area, where malaria is known to be endemic [34], and other vector-borne diseases such as trypanosomiasis [35]. Drinking raw milk, consumption of raw meat, and slaughtering animals at home for human consumption are common practices among more than half of the respondents. This can be attributed to the lack of infrastructure or lack of a structured and lucrative market in the area as reported by more than 95% of the respondents as an important issue. This can also be the continuation of historical and cultural habits. Nevertheless, persisting in such habits reveals the lack of awareness amongst the farmers about the mode of zoonotic diseases transmission through eating habits and management practices. These practices have been identified as contributing factors to zoonotic disease transmission among humans [29]. Similar to our findings, studies among herdsmen and livestock farmers in Zimbabwe [36] and Nigeria [37] reported eating raw meat, drinking raw milk and slaughtering animals at home to be common practices. In contrast to this, other study in Uganda found no consumption of raw milk despite previous historical consumption [38,39]. The need for training and awareness of these practices to equip the farmers on how to safeguard themselves cannot be overemphasized.

In this study, some of the respondents practice eating meat from dead animals instead of burying or burning them. This practice is similar to studies conducted in pastoral communities of Baringo county and in the northeastern part of Kenya where the respondents reported that they prefer to eat carcasses than bury them, they believe that cooking the meat kills all germs [40,41]. It has been postulated that the practice of slaughtering diseased animals for consumption, contributes to outbreaks of zoonoses such as Anthrax, rabies, brucellosis, and Rift Valley fever in rural small-scale livestock farmers [42,43]. Respondents also reported eating raw or undercooked carcasses. This habit (eating raw meat or offal) is a cultural practice among Zulu men, especially during ceremonies. The culture of practicing the consumption of raw meat has been reported in many social groups among farmers in Ethiopia [44], and has been associated with public health problems [45]. 

Many farmers reported that they graze their livestock across the borders of Mozambique and Eswatini, this practice is illegal, particularly within the FMD-controlled zone. Seasonal drought and sociocultural conditions such as animal trade and dowry payments are the major reasons livestock farmers from this region drive their animals across borders. Other factors such as stock theft, dry season, and depleting pastures are the reasons for livestock farmers and their herds crossing borders [46]. It is widely recognized that both within and cross-border animal movements have been attributed to the introduction of new diseases [47]. The development of water boreholes in the communal grazing areas, the adoption of rotational grazing, and the establishment of cattle markets in some of the communities will all go a long way in mitigating the challenges associated with cross-border grazing. It will limit the risks of zoonotic disease transmission and decrease livestock contact with wildlife at watering points. [33].

This study has shown that respondents who do not practice reporting animal diseases and keep animals adjacent to their houses are associated with low knowledge of animal disease. Almost half of the respondents do not practice the quarantine of new animals before they are introduced into the herd, or the communal grazing land and they do not separate diseased animals from their herd. This finding is in line with a previous study conducted in Iran where more than half of the livestock farmers rarely practiced quarantine [48]. This lack of biosecurity practice is a means of introducing new infectious diseases into a healthy herd and communal livestock.

According to respondents’ individual opinions and findings from the PE survey, one of the most prevalent disease conditions include the cases of neurological diseases characterized by cyclic movement in goats, which may be due to tape-worm infestation; whereas, conditions such as coughing may be caused by bovine tuberculosis. In line with our findings, ref. [49] reported respiratory and neurological conditions in livestock as the common disease in Tanzania and are a threat to the livelihood of rural livestock farmers. 

In this study, the respondents identified abortion and malnutrition as fairly common conditions seen in their livestock. Although it is not known what causes abortion in this area, it will be very important to carry out a detailed further study to identify the causes. Recently another study in northern Tanzania reported that very little is known about the infectious causes of abortion in Africa, especially in rural livestock farming communities that depend on livestock for food, income, and wellbeing [50]. It is important that abortogenic diseases, such as Rift Valley fever (RVF) and Brucellosis surveillance, and control should be made a top priority among rural livestock farming communities to achieve better animal production, public health outcomes, and address the knowledge gaps around the causes of abortion in the study area by multifaceted approaches. 

Malnutrition might be due to drought as this is mostly seen during the dry season or syndrome of chronic or sub-chronic diseases. Mastitis was reported as one of the common livestock disease conditions seen every three months, although we do not know the exact causes, it could be attributed to tick bites as this area is endemic to tick infestation [51]. In contrast, a study in Kenya reported diarrhea and respiratory illness as the most common livestock disease syndrome observed [52]; similarly, a study conducted in Sothern Sudan reported diarrhea as the most important and common disease syndrome of livestock [53]. Another study conducted in Colorado among livestock entering the auction market for a one-month period reported the common disease syndrome observed as upper respiratory tract disease, followed by malnutrition and lameness. The least disease syndrome they observed were non-injury- and injury-related hemorrhage and sudden death [54]. 

The majority of the respondents mentioned tick infestation as the most frequent disease condition seen every week. Despite the dipping program in the study area that is managed by the provincial government and the local livestock association, ticks and tick-borne diseases remain a major problem in rural KZN [55], this could be because the ticks have developed resistance to the acaricide [56] or the farmers lack the knowledge of effective use of the acaricides in the dip tanks as prescribed by the manufacturer. 

Most of the respondents mentioned the lack of water, insufficient grazing land, animal theft, the restriction of animal movement, and animal diseases as extremely important challenges faced. This is similar to the findings of a study in Limpopo Province where livestock theft and the lack of a livestock market were reported as some of the major challenges small-scale livestock farmers faced [57]. A study in the Eastern Cape also reported insufficient grazing land and a lack of livestock markets as limiting factors emerging farmers face [58]. In Namibia, the lack of water, animal diseases, and livestock theft were reported as challenges facing livestock farmers [59]. In Zimbabwe, animal diseases, the lack of water, insufficient grazing land, lack of access to veterinary services, cost of veterinary drugs, and livestock theft are major constraints faced by the livestock farmers [60]. Similarly, in DR Congo, animal diseases and the lack of water were the major challenges faced by livestock farmers [61]. To mitigate these challenges and improve livestock production in this study area, there is a need for policies that will focus on improving veterinary and extension services and deployment of resources that will focus on training, capacity building of the rural farmers on animal diseases and husbandry management. There is a need for policies that will focus on the provision of water dams and establishment of rotational grazing areas, especially during thedry season in communal areas. An illegal animal movement task force will be essential in mitigating stock theft and the spread of animal disease, especially at the borders with the neighboring countries. The government should significantly increase the market participation of rural farmers by fostering group marketing.

AHTs are knowledgeable of zoonotic diseases [36] and are the most important source of animal disease information to the farmers and preferred means of communication, followed by community meetings and radio. This is similar to the study conducted in Jordan highlighting the role of radio and veterinarians in raising awareness of zoonotic diseases among livestock farmers [62]. For these reasons, the appropriate medium for dissemination of knowledge will be via the AHTs, community meetings, and radio stations in the isiZulu language. Our findings showed that equipping the AHTs with the appropriate training and skills will go a long way in bridging the knowledge gaps identified especially the lack of animal disease knowledge among young farmers and respondents with formal education in the study area. There is a need for agricultural-related workshops in high schools and communities to stimulate the interest of young pupils in livestock farming. Extension services that will focus on the challenges caused by the common disease conditions identified by the respondent will be essential in mitigating the challenges associated with them. 

## 5. Conclusions and Recommendations

We found that older farmers and those with little or no formal education had more knowledge of animal diseases than younger respondents and those with formal education. Ticks, weakness, weight loss, and worms were the most frequently observed disease conditions. 

The shortage of water and grazing area to increase production, animal theft, restriction of animal movement, animal diseases, and poor access to veterinary services, were cited as the most important challenges faced by the farmers.

The findings demonstrate that there was a lack of knowledge amongst farmers on the risks of contracting zoonoses through the consumption of raw milk, raw meat, and carcasses, and slaughtering animals at home for human consumption. Given the lack of primary health care programs that focus on community awareness of zoonoses and reduction in public health risks at this interface, we highlight the need for multi-disciplinary health campaigns and community sensitization on zoonoses transmission through the primary health workers, the AHTs, community meetings, and radio stations in the isiZulu language. A lot more research needs to be done on the optimal way of managing livestock on communal land and its need to consider all aspects, including socioeconomic.

## Figures and Tables

**Figure 1 ijerph-20-00511-f001:**
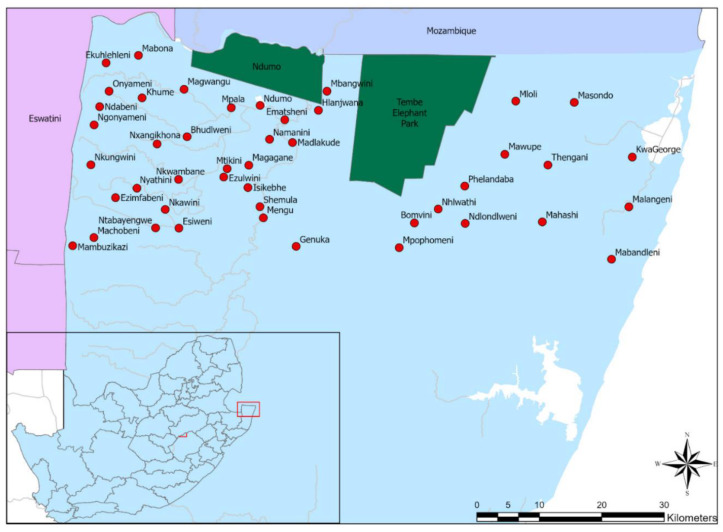
Map of the northern parts of KwaZulu-Natal, South Africa: Dark green is game reserves, red circles are the dip tanks, white color are pans, and grey lines are rivers included in the study.

**Figure 2 ijerph-20-00511-f002:**
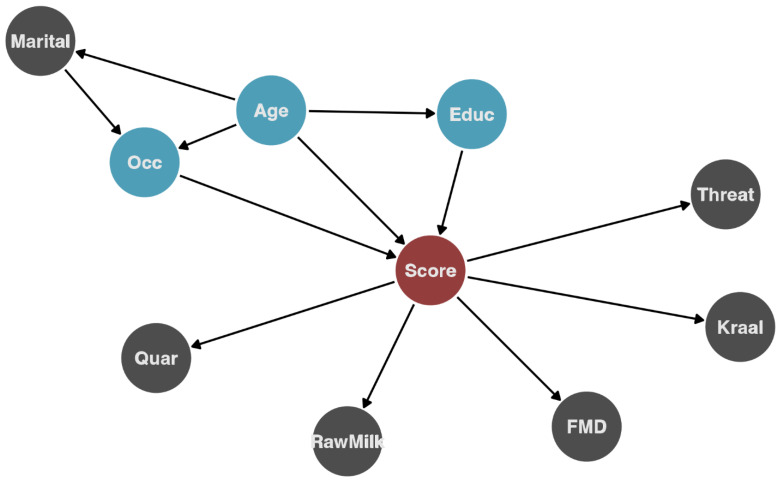
Hypothesized causal pathway for influential variables identified by random forest and their relationship with the outcome (knowledge score). The red circle indicates the outcome (knowledge score), blue circles (which were then included in the regression analysis) are variables hypothesized to directly affect the outcome, and gray for other variables. Marital = marital status; Quar = quarantine practiced; Rawmilk = consumes raw milk; FMD = agrees with the FMD animal movement policy; threat = consider that animal diseases present a threat to livelihood; Kraal = practice kraaling(housing) livestock adjacent to their house; Educ = education level; Age = age group; Occ = occupation.

**Figure 3 ijerph-20-00511-f003:**
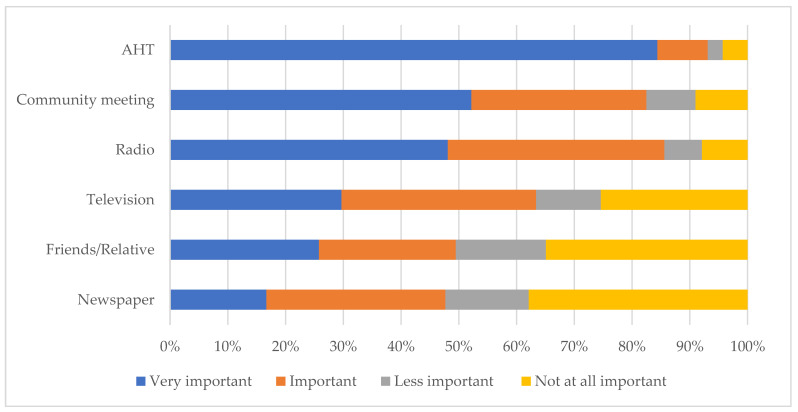
Ranking of the sources of animal disease information and preferred means for communication, in order of importance, amongst communal livestock farmers in northern KwaZulu-Natal.

**Figure 4 ijerph-20-00511-f004:**
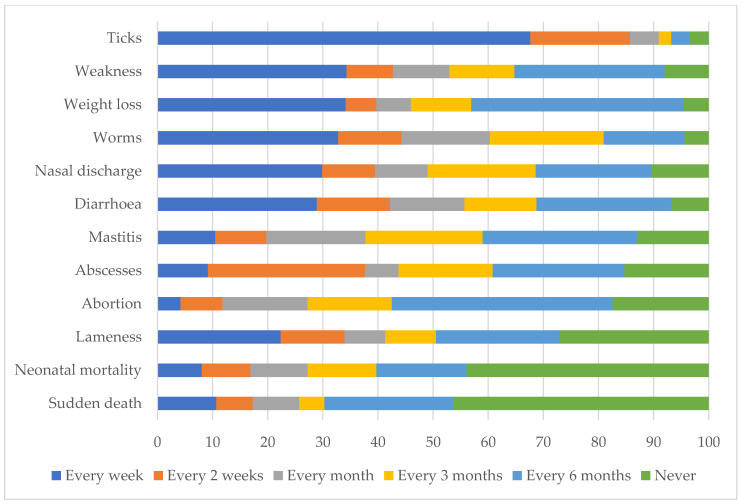
Self-reported frequency of observing livestock disease conditions by communal farmers in far northern KwaZulu-Natal.

**Figure 5 ijerph-20-00511-f005:**
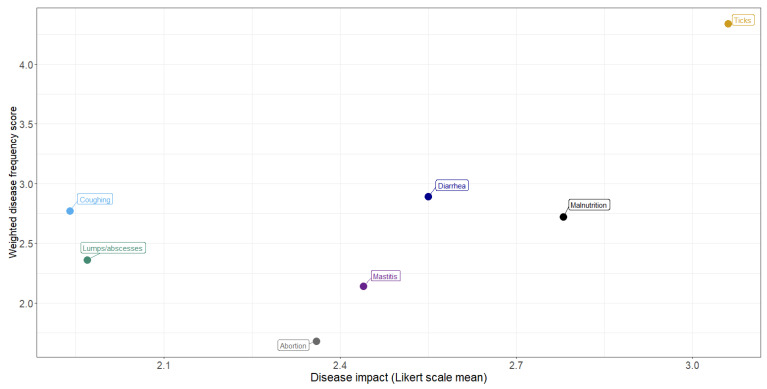
Relationship between frequency and impact of livestock disease conditions: perceptions of communal farmers in northern KwaZulu-Natal.

**Figure 6 ijerph-20-00511-f006:**
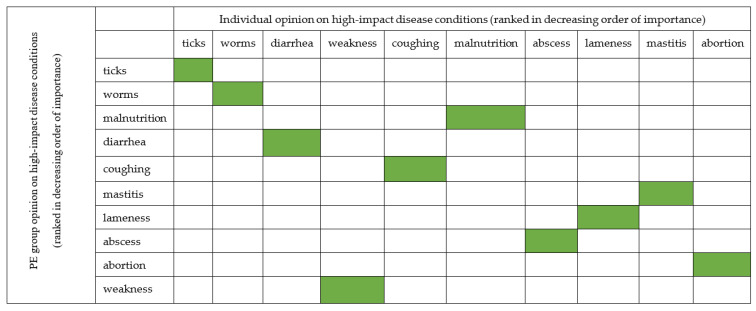
The relationship between individual and PE group survey opinions of Livestock farmers on common diseases condition with high impact in far northern KwaZulu-Natal. The green color indicates the agreement between opinion of the two groups.

**Figure 7 ijerph-20-00511-f007:**
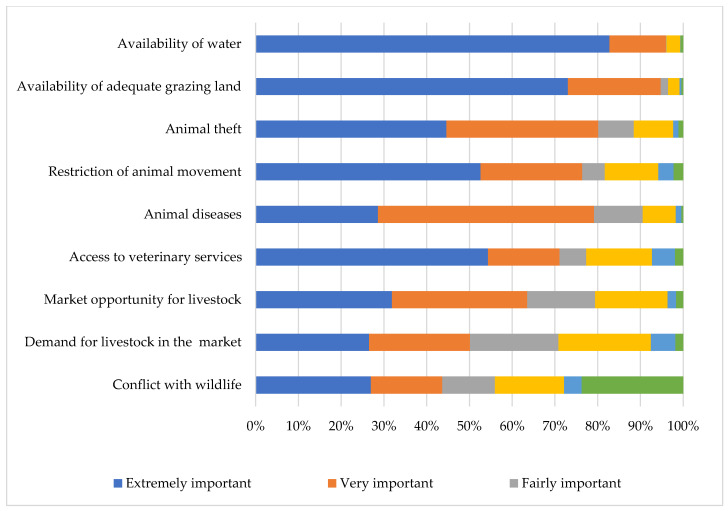
Challenges faced by communal livestock farmers in far northern KwaZulu-Natal.

**Figure 8 ijerph-20-00511-f008:**
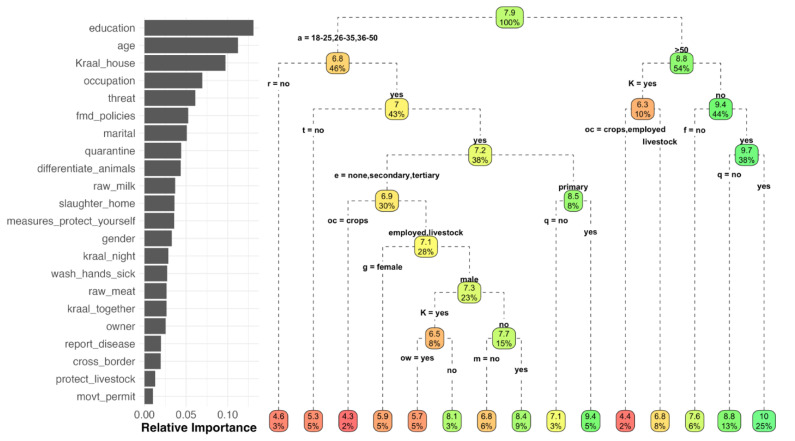
Relative importance of the variables included in the analysis for prediction of the overall knowledge score, and the classification tree, estimated by Random Forest. a = age, r = raw milk, K = kraal, e = education, t = threat, oc = occupation, g = gender, q = quarantine, f = FMD policies, ow = ownership, m = marital.

**Table 1 ijerph-20-00511-t001:** Demographic variables of communal livestock farmers interviewed in far northern KwaZulu-Natal (N = 504).

Variable	Category	Number	Percentage
Age	18–25	29	5.8
	26–35	56	11.1
	36–50	152	30.2
	51–70	246	48.8
	>70	21	4.2
Gender	Female	88	17.0
	Male	416	82.5
Marital status	Married	247	49.0
	Single	250	49.6
	Widow(er)	7	1.4
Education level	None	160	31.7
	Primary	145	28.8
	Secondary	167	33.1
	Tertiary	32	6.4
Occupation	Crop farming	25	5.0
	Employed	27	5.4
	Livestock rearing	452	89.7

**Table 2 ijerph-20-00511-t002:** Demographic information on the animals reared by communal livestock farmers interviewed in far northern KwaZulu-Natal.

Species	Number of Respondents	Total Number of Animals	Mean Number per Household (Median; IQR)
Cattle	485 (96.2%)	9554	19.7 (16; 9–25)
Goats	377 (74.8%)	7256	19.3 (15; 9–22)
Sheep	19 (3.8%)	231	12.2 (8; 4–16)
Pigs	63 (12.5%)	246	3.9 (2; 2–4)
Poultry	414 (82.1%)	9524	23.0 (20; 10–30)
Dogs	201 (39.9%)	501	2.5 (2; 1–3)
Cats	121 (24.0%)	193	1.6 (1; 1–2)

**Table 3 ijerph-20-00511-t003:** Communal livestock Farmers’ knowledge regarding zoonotic diseases transmission and prevention in far northern KwaZulu-Natal.

Question	Number of Respondents	% of Farmers Answering “Yes”	95% CI
1. Are animal diseases preventable?	497	83.5	80.2–87.0
2. Can you get a disease from animals?	500	79.4	75.8–82.9
3. Can you get a disease from a mosquito bite?	499	92.6	90.3–94.9
4. Can you get a disease from the consumption of raw milk?	499	69.7	65.7–73.8
5. Can you get a disease from the consumption of meat from a dead animal?	501	80.8	77.4–84.3
6. Can you get a disease from eating raw or undercooked meat?	501	75.1	71.2–78.6
7. Can you get diseases from touching or consumption of animal blood?	500	66.8	62.7–70.9
8. Can you get a disease from handling aborted fetuses?	500	65.2	61.0–69.4
9. Can you get a disease from keeping animals adjacent to your house?	501	59.3	55.0–63.6
10. Can livestock get a disease from wildlife?	501	78.4	74.8–82.1
11. Can wildlife get a disease from livestock?	476	53.4	48.9–57.9

**Table 4 ijerph-20-00511-t004:** Communal farmers’ practices relating to prevention and management of zoonotic and animal diseases in far northern KwaZulu-Natal.

Variable	Number of Respondents	% of Farmers Answering “Yes”	95% CI
1. Do you consume raw milk?	501	52.3	47.9–56.7
2. Do you eat raw meat?	497	24.8	20.9–28.6
3. Do you wash hands after touching sick or dead animals?	501	98.2	97.0–99.4
4. Do you report animal diseases?	504	94.6	92.7–96.6
5. Do you use protective gear to protect yourself when your animals are sick?	435	71.0	66.8–75.3
6. Do you use prophylactic treatment to protect your livestock?	488	95.5	93.6–97.3
7. Do you obtain a movement permit before you move cattle or goat into or out of your community?	500	96.8	95.3–98.3
8. Do your animals cross the Mozambique or Eswatini border to graze?	503	13.9	10.9–17.0
9. Can you differentiate animals that are from neighboring countries and those from South Africa?	499	57.1	52.6–61.5
10. Do you quarantine new animals before introducing them into your herd?	501	51.5	47.1–55.9
11. Do you kraal your animals at night?	497	92.0	89.6–94.4
12. Do you kraal different livestock species together?	498	13.1	10.1–16.0
13. Do you kraal your livestock adjacent to your house?	474	21.9	18.2–25.7
14. Do you slaughter animals at home for human consumption?	497	87.1	84.2–90.1

**Table 5 ijerph-20-00511-t005:** Levels of knowledge, attitudes, and practices of communal livestock farmers regarding livestock diseases in far northern KwaZulu-Natal.

Levels	Score	Frequency	Percentage (%)	95% CI
Knowledge regarding zoonotic diseases and transmission (overall total score is 11)				
High level	9–11	268	53.2	48.8–57.5
Moderate level	6–8	128	25.4	21.6–29.2
Low level	0–5	108	21.4	17.8–25.0
Attitude of livestock farmers regarding the impact of diseases on their financial income				
Do you consider animal diseases as a threat to your financial income?				
Yes	1	457	91.6	89.1–94.0
No	0	42	8.4	6.0–10.8
Attitude regarding FMD animal movement policy				
The current animal movement control policy is the best way to control the spread of FMD?				
Agree	1	408	81.6	78.2–85.0
Disagree	0	92	18.4	15.0–21.8
Practice regarding disease prevention				
Very good	3	171	33.9	3.0–38.2
Good	2	129	25.6	2.0–29.6
Poor	0–1	204	40.5	36.3–44.8
Practice regarding handling of sick animals				
Very good	3	234	46.4	42.1–50.8
Good	2	254	50.4	46.0–54.8
Poor	0–1	16	3.2	2.0–5.3
Practice regarding carcass disposal				
Very good	3	23	4.5	3.0–6.8
Good	2	347	68.9	64.7–72.8
Poor	0–1	134	26.6	22.9–30.3

**Table 6 ijerph-20-00511-t006:** Farmers’ perception of the impact of disease conditions on their livestock.

Disease Conditions	Weighted Disease Frequency Score	Likert Scale Mean of Impact of Disease
Ticks	4.34	3.06
Malnutrition	2.72	2.78
Diarrhea	2.89	2.55
Abortion	1.68	2.36
Mastitis	2.14	2.44
Lumps/abscesses	2.36	1.97
Coughing	2.77	1.94

**Table 7 ijerph-20-00511-t007:** Individual and PE opinions of Livestock farmers on high-impact diseases condition in far northern KwaZulu-Natal ranking (Z–A).

Individual Survey			PE Group Survey		
Name of Disease Condions	Freq.	Weighted Disease Frequency Score	Name of Disease Condions	Freq. of Stones	Average Proportion Pilling Score
ticks	466	4.34	ticks	1058	21.81
worms	436	3.11	worms	857	17.67
diarrhea	462	2.89	malnutrition	689	14.21
weakness	383	2.87	diarrhea	484	9.98
coughing	410	2.77	coughing	420	8.66
malnutrition	446	2.72	mastitis	385	7.94
abscess	400	2.36	lameness	334	6.89
lameness	380	2.21	abscess	331	6.82
mastitis	409	2.14	abortion	242	4.99
abortion	412	1.68	weakness	50	1.03

**Table 8 ijerph-20-00511-t008:** Univariable associations of variables with knowledge score amongst communal livestock farmers in northern KwaZulu-Nata.

Variable	Number of Respondents	Percentage	Estimate	95% Cl	*p*-Value
**Education**					
None *	160	31.8			
Primary	145	28.7	−0.161	(−0.77, 0.45)	0.603
Secondary	167	33.1	−1.937	(−2.52, −1.35)	<0.001
Tertiary	32	6.4	−2.163	(−3.19, −1.14)	<0.001
**Age**					
>50 *	267	52.9			
36–50	152	30.2	−1.573	(−2.11, −1.04)	<0.001
26–35	56	11.1	−2.374	(−3.15, −1.60)	<0.001
18–25	29	5.8	−2.705	(−3.73, −1.68)	<0.001
**Marital Status**					
Married	247	49.0			
Single	250	49.6	−1.070	(−1.56, −0.58)	<0.001
Widow	7	1.3	−1.640	(−3.75, 0.46)	0.126
**Kraal (house)**					
No *	370	78.1			
Yes	104	21.9	−1.975	(−2.58, −1.37)	<0.001
**Occupation**					
Crop farming *	25	5.0			
Employed	28	5.6	1.646	(0.14, 3.15)	0.032
Livestock rearing	451	89.4	2.633	(1.51, 3.76)	<0.001
**Threat**					
No *	42	8.4			
Yes	457	91.6	2.542	(1.67, 3.42)	<0.001
**FMD Policies**					
No *	92	18.4			
Yes	408	81.6	1.996	(1.38, 2.61)	<0.001
**Quarantine**					
No *	243	48.5			
Yes	258	51.5	1.333	(0.85, 1.82)	<0.001
**Raw Milk**					
No *	239	47.42			
Yes	262	51.98	−0.540	(−1.30, −0.04)	0.032

* Reference category, CI = confidence interval, Missing values for FMD policies = 4, Missing values for Threat = 5, missing values for quarantine = 3, missing values for Kraal = 30, Missing values for Differentiate Animals = 5, Missing values for Raw Milk = 4.

**Table 9 ijerph-20-00511-t009:** Factors associated with knowledge score amongst communal livestock farmers in northern KwaZulu-Natal: multivariable regression model.

Variable	Estimate	95% CI	*p*-Value
**Education**			
None *	-		
Primary	−0.089	(−0.67, 0.49)	0.765
Secondary	−1.135	(−1.77, −0.49)	0.001
Tertiary	−1.379	(−2.43, −0.32)	0.010
**Age**			
>50 *	-		
36–50	−1.202	(−1.75, −0.64)	<0.001
26–35	−1.546	(−2.41, −0.67)	<0.001
18–25	−2.055	(−3.20, −0.90)	<0.001
**Occupation**			
Crop farming *	-		
Employed	1.783	(0.35, 3.20)	0.014
Livestock rearing	2.686	(1.63, 3.72)	<0.001

* Reference category.

## Data Availability

All the data are contained within the article.

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
