# Peer review of "Knowledge, Attitudes, and Practices of Communal Livestock Farmers regarding Animal Health and Zoonoses in Far Northern KwaZulu-Natal, South Africa"

_ijerph, 2022, doi:10.3390/ijerph20010511_

Round 1

Reviewer 1 Report

The authors performed an interesting questionnaire-based cross-sectional survey on the knowledge, attitudes and practices, as well as challenges faced by communal farmers regarding livestock and zoonotic diseases in a wildlife- livestock interface area in KwaZulu South Africa. The study revealed an overall low knowledge about wildlife and livestock zoonotic diseases and poor disease prevention behaviours among 504 farmers interviewed at 45 dip tanks. Results of the study are important for farmers, stakeholders and policymakers,  in that they highlight the need for more education to improve farmer’s knowledge and promote disease prevention practice, drive attention to the several challenges faced regarding animal production and propose concrete strategies to mitigate these problems. The manuscript is written in a fluent way, is easy to follow and well-structured.

Below some minor comments:

Line 20: please check the grammar of the sentence

Line 128-129: word at the beginning of the sentence missing.

Here and along the manuscript it is not always clear to me which questionnaires were administered to individuals and which were group interviews (PE). Were the same people interviewed individually at each dip tank also interviewed as a group? Could you please clarify the two questionnaires where applicable? Results presented in tables and figures are all, with the exception of table 7 and figure 6 from individual interviews? A short explanation why the authors chose this approach (what is the advantage) would be helpful for better understanding the interview procedure.

2.5 Questionnaires, lines 152-169

In the results section I could clearly follow the questionnaire outputs from sections I-VII of the survey, which are briefly explained in lines 144-151. Here however, it is more difficult to attribute the information provided to the different sections.  For instances, I understood that questions about the challenges faced by farmers belong to section VII, but here it appears before the heading 2.5.1 Farmers’ knowledge regarding zoonotic disease transmission which belongs to section II (?).

Lines 186-187: Is “Yes” really the best response in respect to disease prevention for all the questions listed in table 4? For instances, consumption of raw milk and raw meat, home slaughtering?

Lines 189-194: I believe the practices mentioned in these lines (protective gear and prophylactic treatments) correspond to practices 5 and 6 of table 4 (?). Could you please reformulate slightly to make it more comprehensible?  For instances by adding “Do you use protective gear to protect yourself when your animals are sick”, and “Do you use prophylactic treatment to protect your livestock”?

Lines 195-199: not very clear to me where to retrieve the mentioned five measures

Lines 296-299: Please check percentages provided in table 5 under “Practices regarding zoonotic diseases and management”

Lines 306-307: according to table 5 it should state: (…) 4.5% (95%CI (…) had very good practice.

Discussion

I am not native English speaker, but there seem to be some minor grammar/sentence issues along the Discussion section, please review carefully. Below some of the writing issues detected:

Line 461-466: please check the structure/grammar of the sentence, sentence can be split, e.g., (…). This is similar to findings in a (…)

Line 466: Similarly, in a (…) that assessed (…)

Line 470-472: please check the structure/grammar of the sentence

Lines 527-528: please correct to (…) slaughtering diseased animals (…)

Line 530: please correct to: Respondents also reported (…)

Line 547: please correct to:  (…) respondents who don’t practice (…)

Line 584: please check: (….) the most (?) frequent (…)

Lines 638-640: please delete these sentences

Author Response

Dear reviewer,

Thank you for the time you took to review our manuscript, attached is our response to your valuable comments.

Reviewer 2 Report

The manuscript presents interesting data on the knowledge, attitudes and practices of communal livestock farmers in far northern KwaZulu-Natal (KZN) province, South Africa, in regard to livestock and zoonotic diseases found at the livestock-wildlife interface and to determine the constraints and challenges faced by communal livestock farmers. In my opinion, the manuscript is well-written and the investigations are well-planned and performed.

Author Response

Dear reviewer,

Thank you for the time you took to review our manuscript.

Reviewer 3 Report

Comments for authors:

Title:

 Knowledge, attitudes and practices……to be  Knowledge, attitudes, and practices….

Abstract:

·         L22:   ……. dip…….  Please write the full name of the abbreviation

·        L29: compared to younger farmers…..  to be  compared to younger ones.

·        In general, Please try not repeat the same words too much either in the same sentence or the next one.

Introduction:

·        L39:  both rural areas [1-3] and urban areas [4]. Delete the repeated word areas.

·        Please do not use many references for the same information or paragraph. Only, keep the most recent one specially that the list of references are too much

Materials and methods:

Study area:

·        L 82-83: Kindly add the longitude and altitude of the mentioned places as most foreigner readers can’t know such areas.

·        L89 :  “ ethnic group” ………. Please do not try the same words many times. Kindly, take in consideration across all the manuscript.

·        L100: AHTs: Please write the full name of the abbreviation for the first time in manuscript, Kindly, take in consideration across all the manuscript.

·        L129: …..at each dip tank. Each dip tank……. Please do not try the same words many times. Kindly, take in consideration across all the manuscript.

·        L185: …..were administered ……..were delivered….. 

Results:

 L254: more explanation of data from table (3) should be added.

Discussion:

·        L452-466:  very long paragraph with only one cited reference. Kindly make it shorter and no need to repeat the same results information.

·        L473-485:  This paragraph is more suitable to be merged with section 3.9. in the M.M because it is only explaining the analysis methods, variables, ……etc. not discuss the obtained results.

·         L497: You cited 5 references to explain some ideas and left many paragraphs without references. Please decrease this large number and take in consideration across the whole manuscript.

·        L505:  The findings of this study show………..The findings of this study showed…….. Please make the tense of all sentences across the manuscript in the past time.

·        L505-507: 3 references were used please decrease as requested before.

·        L509: ………….is common practice………are common practices……

·        L516-519: The same problem again 4 references in one paragraph.

·        L532-534: Please use only one reference.

·        L540: Kindly, use only one reference.

·        L555-557: Please do not repeat the results as it is in the discussion part

References:

 Using of 83 cited references is too much for a research manuscript not a review one Although you have left many long paragraphs without references. Kindly, downsize this huge number  through deleting the old ones specially for those being used in the same paragraph.

Author Response

Dear reviewer,

Thank you for the time you took to review our manuscript, attached is our response to your valuable comments as updated in the manuscript in tract changes.
